# Sulfur Exafluoride Contrast-Enhanced Ultrasound Showing Early Wash-Out of Marked Degree Identifies Lymphoma Invasion of Spleen with Excellent Diagnostic Accuracy: A Monocentric Study of 260 Splenic Nodules

**DOI:** 10.3390/cancers14081927

**Published:** 2022-04-11

**Authors:** Marco Picardi, Claudia Giordano, Fabio Trastulli, Aldo Leone, Roberta Della Pepa, Novella Pugliese, Rossella Iula, Giuseppe Delle Cave, Maria Gabriella Rascato, Maria Esposito, Elena Vigliar, Giancarlo Troncone, Massimo Mascolo, Daniela Russo, Marcello Persico, Fabrizio Pane

**Affiliations:** 1Department of Clinical Medicine and Surgery, Federico II University Medical School, Via Sergio Pansini 5, 80131 Naples, Italy; marco.picardi@unina.it (M.P.); fabio.trastulli@unina.it (F.T.); aldo.leone93@libero.it (A.L.); robertadellapepa@unina.it (R.D.P.); novypugliese@yahoo.it (N.P.); rossella.iula@gmail.com (R.I.); giusedellecave87@gmail.com (G.D.C.); mg.rascato@gmail.com (M.G.R.); azzura92me@gmail.com (M.E.); fabrizio.pane@unina.it (F.P.); 2Department of Advanced Biomedical Sciences, Federico II University Medical School Naples, Via Sergio Pansini 5, 80131 Naples, Italy; elena.vigliar@unina.it (E.V.); giancarlotroncone@unina.it (G.T.); 3Department of Public Health, Federico II University Medical School Naples, Via Sergio Pansini 5, 80131 Naples, Italy; massimo.mascolo@unina.it (M.M.); daniela.russo@unina.it (D.R.); 4Department of General Surgery, Endocrinology, Orthopaedics, and Rehabilitations, Federico II University Medical School Naples, Via Sergio Pansini 5, 80131 Naples, Italy; marcellopersico@hotmail.com

**Keywords:** lymphoma, contrast enhanced ultrasonography, spleen nodules

## Abstract

**Simple Summary:**

Our retrospective collection from the database registry of the Hematology Unit of the Federico II University Medical School of Naples (Italy) of all patients referring to our center (from 1 January 2009 to 31 January 2019) for lymph node biopsy-proven lymphoma and one or more distinct splenic lesions, visible at baseline ultrasonographic scans and submitted to CEUS, could be a great asset in the diagnostic approach for splenic lymphoma. For the first time, based on a robust sample size of 260 nodules (with final diagnoses uniformly controlled by clinical laboratory imaging follow-up, in the cases not directly biopsy-proven) we showed that CEUS can achieve the diagnosis of splenic malignant lymphoma with excellent accuracy.

**Abstract:**

Contrast-enhanced ultrasonography (CEUS) use for detecting lymphoma in the spleen was questioned because of the risk of its inadequate diagnostic accuracy. The aim of the present study was to validate CEUS exam for the identification of spleen involvement by lymphoma in patients at risk. A total of 260 nodules from the spleens of 77 patients with lymph node biopsy-proven non-Hodgkin lymphoma (NHL; *n* = 44) or Hodgkin lymphoma (HL; *n* = 33) at staging (*n* = 56) or follow-up (*n* = 21) were collected in a hematology Italian center and retrospectively analyzed. Nodules were classified as malignant lymphoma if ≥0.5 cm (long axis) with arterial phase isoen-hancement and early (onset <60 s after contrast agent injection) wash-out of marked (≤120 s after contrast agent injection) degree. Other perfusional combinations at CEUS scans qualified lesions as benign or inconclusive. Diagnostic reference standard was clinical laboratory imaging monitoring for 230 nodules, and/or histology for 30 nodules. The median nodule size was 1.5 cm (range 0.5–7 cm). According to the reference standard, 204 (78%) nodules were lymphomas (aggressive-NHL (a-NHL), 122; classic-HL (c-HL), 65; indolent (i)-NHL, 17) and 56 (22%) were benign (inflammation, infection, and/or mesenchymal) lesions. Sensitivity, specificity, positive predictive value, negative predictive value, and overall diagnostic accuracy of CEUS for detecting lymphoma in the spleen were 95%, 100%, 100%, 85%, and 96%, respectively. Marked wash-out range of 55–90 s (median, 74 s), 92–120 s (median, 100 s), and 101–120 s (median, 114.5 s) was 100%, 96.6%, and 77% predictive of a-NHL, c-HL, and i-NHL splenic nodular infiltration, respectively. The CEUS perfusional pattern of arterial phase isoenhancement with early wash-out of marked degree was highly accurate for the detection of lymphomatous invasion of spleen in patients at risk, enabling its use for a confident non-invasive diagnosis.

## 1. Introduction

Spleen invasion in lymphoproliferative malignancies occurs in a significant proportion of patients: involvement is reported in up to 40% of patients with non-Hodgkin lymphoma (NHL) and Hodgkin lymphoma (HL), and this increases to 70% at postmortem [1]. Although histologic evidence is the reference standard for the diagnosis of involvement by lymphoma, for obvious practical and ethical reasons, it is not possible to perform biopsies in all organs of interest, especially the spleen [2]. A growing number of clinical trials has well-established the accuracy of conventional imaging tools in detecting lymphomatous nodules in the spleen. By analyzing the data, the pooled summary sensitivity rate was 85% (range 50–100%) for 18F-fluoro-deoxy-glucose (FDG)-positron emission tomography (PET), 85% (range 62–96%) for contrast-enhanced magnetic resonance imaging (MRI), and 58% (range 33–94%) for contrast-enhanced computed tomography (CT) [1,3,4,5,6,7]. By contrast, modern ultrasonographic technology requires validation in a large series of splenic nodules at risk of involvement by lymphoma [6].

The American College of Radiology has recently released a very useful system (Contrast-Enhanced Ultrasonography (CEUS)/Liver Imaging Reporting and Data System (LI-RADS)) to classify nodules at risk of hepatocellular carcinoma (HCC) investigated by CEUS [8]. The LI-RADS classification system includes a category of lesions with the typical diagnostic pattern of HCC, termed LR-5, which corresponds to arterial phase hyperenhancement followed by late and mild degree of wash-out. Another category of lesions with malignant features, named LR-M, which is suggestive of the possibility of non-hepatocellular malignancies, includes the presence of arterial phase rim enhancement or early onset of wash-out of marked degree [9]. The numerous benefits of CEUS, compared to its risks, have recently made it an emergent imaging technique for non-hepatic applications as well, specifically in the evaluation of splenic lymphoma [10]. Although of great potential interest, CEUS is considered as a supplement to cross-sectional imaging in defining lymphoma in the spleen according to recent recommendations for lymphoma assessment [11,12]. Furthermore, reports on CEUS regarding its application in the most representative subtypes of lymphomas affecting the spleen and possible diagnostic criteria in establishing involvement by these different subtypes are scanty.

The aim of this study was to test the accuracy of intravenous microbubble-based microvasculature ultrasonographic assessment of sulfur exafluoride for diagnosing lymphoma nodular infiltration of the spleen. In addition, we explored the CEUS perfusional patterns of the three principal clinical categories of lymphomas, i.e., aggressive NHL (a-NHL), classic (c)-HL, and indolent NHL (i-NHL), affecting the spleen. Finally, we examined how CEUS results impacted the diagnostic flow-chart of spleen status in patients with lymphoma at risk of splenic involvement. 

## 2. Materials and Methods

### 2.1. Study Design

In our Institution, CEUS is held as a routine imaging practice for all significant lesions detected in the spleen at risk for lymphomatous invasion [6,13]. Herein, this retrospective collection from the database registry of the Hematology Unit of the Federico II University Medical School of Naples (Italy) reflected an almost consecutive enrolment of all subjects referred to our center for lymph node biopsy-proven lymphoma (by excisional biopsy or imaging-guided core-needle biopsy (CNB)) [14,15,16] and clinical suspicion of spleen involvement for one or more distinct splenic lesions, which were visible at baseline ultrasonographic scans and thereafter submitted to CEUS. After checking CEUS reports, images, and/or video clips from the Hematology Unit database of the Federico II University of Naples (Italy), we acquired more detailed information on splenic lesions suspected of lymphoma, i.e., the precise determination of the arterial phase enhancement, the timing of the onset of wash-out appearance of microbubbles, and a judgement as to the degree of contrast intensity at different time points during the venous phase.

All necessary approvals were obtained from the Ethic Committee of the Federico II University Medical School of Naples (Italy). Given the retrospective design, the acquisition of all individual informed consents was waived (except for performing microbubble intravenous infusion at CEUS time).

### 2.2. Participants

Inclusion criteria were: (i) histological diagnosis of lymphoma (revised according to the World Health Organization 2016 classification), which was managed and/or treated according to the National Comprehensive Cancer Network guidelines [14,15,16,17,18]; (ii) baseline ultrasonographic (B-mode, with tissue harmonic compound technology) scans showing solid nodules of spleen, in particular hypoechoic nodular lesions ≥0.5 cm as long axis, and investigated by CEUS; (iii) availability of the CEUS information of nodules reporting the arterial phase pattern, the timing of the onset and degree of wash-out whenever this feature occurred (either reported on the original report or, whenever this was missing, as assessed retrospectively by the investigators on recorded video clips and images); and (iv) validation of CEUS results by a reference standard, i.e., at least one conventional imaging tool (FDG-PET, contrast-enhanced CT, and/or contrast-enhanced MRI performed as already described) [5,6,7], which clearly defined the malignancy or benignity according to already reported features [5,7,19].

The definitive diagnoses of the subtypes of lymphomatous nodules affecting the spleen were established based on histologic results of superficial or deep-seated lymph node biopsies plus clinical/laboratory/imaging follow-up (lasting at least six months after the CEUS examinations (monitoring after therapy)), or whenever available, splenectomy or radiologically guided CNB [20]. The definitive etiologies of benign lesions in the spleen were obtained by clinical/laboratory/imaging follow-up lasting at least six months after the CEUS examinations (monitoring after therapy). 

This retrospective study was conducted in the Hematology Unit of the Federico II University of Naples (Italy).

The patients included in the study underwent CEUS examinations of the spleen from 1 January 2009 to 31 January 2019.

#### 2.2.1. Splenic CEUS Examination Methods

Ultrasonographic investigation of the spleen was obtained by hematologists with extensive expertise in lymphomas management and spleen CEUS (M.P., R.DP., N.P., and F.T. with more than 10 years of experience with lymphoma diagnostic work-up and therapy, and US and CEUS in the field of hematological malignancies) [6,13], using different equipment from the same manufacturer along the entire period of study (iU22, Philips Healthcare, Bothell, Wash, equipped with tissue harmonic compound technology [SonoCT] and a 2–5 [MHz broadband curvilinear probe, from 2009 to 2016; and EPIQ-5, Philips Healthcare, Bothell, Wash, equipped with SonoCT and a 1–5 MHz broadband curvilinear probe, from 2017 to 2019). These scanners were employed for the baseline study, depicting one or more distinct hypoechoic nodules, whose size was determined by measuring the long axis (in centimeters). When one or more possible nodules were detected in the spleen, specific informed consent was obtained for the contrast-enhanced study aimed at characterizing all the nodular lesions. Nodule vascularization was assessed with a contrast agent containing sulfur hexafluoride–filled phospholipids-stabilized microbubbles (Sonovue; Bracco, Milan, Italy) and Philips equipment operating at low acoustic pressure with contrast-specific software. For each patient, a total 2.4 mL dose of contrast agent was rapidly injected through a peripheral vein by using a 20-G needle. Perfusion US of the spleen started immediately after contrast agent injection and lasted 7–9 min, including the arterial phase (starting 10 s after contrast agent injection) and the parenchymal phase (starting 40 s after contrast agent injection, with the progressive appearance of microbubbles in the sinusoidal bed and terminating with bubble disappearance) [6]. 

#### 2.2.2. Splenic CEUS Patterns

Arterial phase isoenhancement was defined as a lesion becoming globally isoechoic compared to the surrounding parenchyma, in the arterial phase of the study. 

Arterial phase hypoenhancement was defined as a lesion becoming globally hypoechoic compared to the surrounding parenchyma, in the arterial phase of the study.

Arterial phase hyperenhancement was defined as a lesion becoming globally or partially hyperechoic (but not with rim or globular peripheral distribution) compared to the surrounding parenchyma, in the arterial phase of the study.

A rim enhancement pattern (not globular peripheral) was defined as increased enhancement mainly concentrated at the periphery of the nodule in the arterial phase of the study, followed by late wash-out.

A peripheral discontinuous globular pattern was defined as peripheral enhancement in the arterial phase with progressive centripetal partial or complete fill-in.

In the arterial phase of the study, a reverse rim sign describes relative hypoenhancement of the periphery of the nodule and normal enhancement in the remaining portion of the lesion.

Wash-out was defined when the lesion became hypoechoic compared to the surrounding parenchyma in the portal-venous phase. When such wash-out occurred, it was further classified according to its timing: as “early” if it appeared before 60 s following contrast injection or as “late” if it occurred after 60 s, and according to its intensity, as “marked” when the lesion become hypoenhanced or dark within 2 min (otherwise defined as “mild”) following contrast injection [6].

#### 2.2.3. Splenic CEUS Test Positivity

NHL- and HL-diagnostic criteria (DC) at CEUS were defined as nodules ≥0.5 cm with hypoenhancement in the parenchymal phase, depicted as a progressive increase of nodule-to-parenchyma contrast gradient (i.e., a clear hypoechoic defect of the lesion, appearing almost punched-out or echo-free) while moving from the arterial phase to the late phase of parenchymal opacification [6]. 

Benign findings (BF)-DC at CEUS comprised nodules ≥0.5 cm with: (i) hypoenhancement in the arterial phase without late wash-out (granulomatous lesions, infarction), (ii) isoenhancement in the arterial phase with or without late wash-out of mild degree intensity (capillary hemangiomas), (iii) discontinuous globular peripheral enhancement followed by progressive centripetal fill-in or very late (>3 min) wash-out of mild degree (cavernous hemangiomas, (iv) a thin rim enhancement pattern in the arterial phase followed by wash-out in the venous phase (abscess), (v) reverse rim sign (granulocyte colony-stimulating factors (G-CSF)-related myeloid metaplasia), and (vi) hyperenhancement in the arterial phase without later wash-out (hamartomas) [10,21].

#### 2.2.4. Immunohistochemical Study of the Tumoral Angiogenesis of Splenic Nodules

In a post hoc assessment, we analyzed the microvessel density (MVD) by immunohistochemical and morphometric evaluations of vascular endothelial growth factor-A (VEGF-A), VEGF receptor (Flk-1), and cyclooxygenase 2 (COX2) expression, as already described by others [22]. This assessment was performed in splenic nodules containing unequivocal B cell lymphoma, as assessed by H&E morphology in combination with immunohistochemistry, for which formalin-fixed, paraffin-embedded samples were available from the archive of the Institute of Pathology at the Medical University of Federico II, Naples (Italy).

### 2.3. Statistical Analysis

Sensitivity (probability that a CEUS result will be positive when the disease is present (true-positive rate)), specificity (probability that a CEUS result will be negative when the disease is not present (true-negative rate)), positive predictive value (PPV; probability that the disease is present when the test is positive), negative predictive value (NPV; probability that the disease is not present when the test is negative), and, lastly, the diagnostic accuracy (overall probability that a nodule is correctly classified = Sensitivity × Prevalence + Specificity × (1 − Prevalence)) [23] were reported.

Continuous variables were reported with their median and range. A descriptive analysis was carried out, reporting rates in absolute numbers and percentages. Comparisons among groups were calculated using nonparametric (Mann–Whitney and Wilcoxon) tests. Categorical variables were considered significant at *p* < 0.05. Analysis of the data was performed using SPSS statistical analysis software (SPSS Inc., Chicago, IL, USA).

## 3. Results

### 3.1. Participants

Data from a total of 264 spleen nodules were collected in 78 patients. One patient with four nodules was excluded for equivocal histologic results regarding specimens obtained by using imaging-guided CNB. Thus, the final analysis was performed on 260 nodules in 77 patients receiving the scheduled dose of contrast agent. The average time required for the complete ultrasonographic assessment was 30 min (range 25–45 min); the perfusional examinations were well-tolerated without complications. All ultrasonographic scans were assessable for the final classification. A consolidated standard for reporting of diagnostic accuracy study (STARD) diagram summarizes the flow of nodules through the study in Figure 1.

Patient and splenic nodule characteristics are reported in Table 1.

The median age of analyzed patients was 48 years (range 22–72 years). Males were 57% of the cases. Fifty-six patients were found to have 198 (76%) nodules in the spleen at the time of the staging process of lymphomas. They were scheduled to receive front-line therapy (Adriamycin, Bleomycin, Vinblastine and Dacarbazine (ABVD) in 30 cases; Rituximab, Cyclophosphamide, Hydroxydaunorubicin, Vincristine, and Prednisone (R-CHOP) in 23 cases; and Rituximab and Bendamustin (R-Benda) in three cases) [18]. Twenty-one patients were found to have 62 (24%) nodules during follow-up for lymphomas previously treated (R-CHOP (*n* = 14 cases), CHOP (*n* = 4 cases), and ABVD (*n* = 3 cases)).

The median size of nodules was 1.5 cm (range 0.5–7.0 cm). Two or more nodules occurred in 47 patients (2–4 nodules in 21 patients; ≥5 nodules in 26 patients), while a single nodule occurred in 30 patients. After clinical and whole-body imaging evaluations, 230 splenic nodules were accompanied by superficial and/or deep-seated lymphadenopathies, whereas the remaining 30 nodules occurred in the absence of other sites of involvement by disease.

#### 3.1.1. Spleen Status According to the Reference Standard

According to the reference standard, a total of 204 (in 55 patients) of 260 nodules were classified as lymphomas: a-NHL for 122 nodules (108, diffuse large B cell lymphoma (DLBCL); 10, T-rich B-cell lymphoma (TRBCL); four, anaplastic T-cell lymphoma (ATCL)); c-HL for 65 nodules (50, nodular sclerosis; 11, mixed cellularity; and four, lymphocyte-rich subtypes); and i-NHL for 17 nodules (grade 1–2 follicular lymphoma [FL]; Table 1). Nodules were largest in a-NHLs (median size: 1.8 cm; range 0.6–7 cm) and smallest in i-NHL (median size: 0.6 cm; range 0.5–0.8 cm). The median size of the nodules of c-HL was 1.5 cm (range 1–2 cm).

The remaining 56 nodules (in 22 patients) were found to be of benign etiology acording to the reference standard (20 granulomatous processes for sarcoidosis, 20 hemangiomas, 10 abscesses fungus, eight nodules; bacteria, two nodules), four G-CSF-related myeloid metaplasias, one hamartoma, and one infarction; Table 1).

#### 3.1.2. Spleen Status According to CEUS

A total of 194 nodules had CEUS test positivity for lymphoma. The enhancement pattern characterized by the iso (in the arterial phase)-hypo (in the venous phase), i.e., isoenhancement in the arterial phase followed by wash-out appearance that had early onset and marked degree, occurred in 100% of these lesions. In particular, the wash-out onset occurred at a median of 43 s (range 20–60 s), and the median timing of the marked wash-out was 74 s (range 55–120 s). The dynamic patterns of the 56 nodules classified as benign according to the CEUS are reported in Table 2, together with the corresponding median size and the specific etiology (as established by the reference standard).

CEUS examinations of the remaining 10 nodules were classified as inconclusive. The median size of such nodules was 1 cm (range 0.8–1.5 cm). All inconclusive nodules appeared with a pattern characterized by arterial phase isoenhancement followed by mild and late wash-out of contrast agent. Figure 1 shows the final diagnoses of CEUS inconclusive nodules, which were all depicted as malignant according to the reference standard. Five nodules were due to c-HL (mixed cellularity, three; nodular sclerosis, two), and five nodules were due to i-NHL (grade 1–2 FL, five).

### 3.2. Diagnostic Accuracy of CEUS

There was excellent concordance in the classification of positive findings for malignancy between the reference standard and CEUS scans. The sensitivity rate of spleen lymphomatous malignant status was 95% (95% confidence interval (CI): 91% to 98%), i.e., 194 of 204 nodules positive for lymphomas were identified by CEUS, with a false negative rate of 5% (10 (those with inconclusive results at CEUS, as reported above) of 204 nodules positive for lymphoma were not identified). Noteworthy was the specificity rate of 100% (95% CI: 94% to 100%), PPV rate was 100% (95% CI: 98% to 100%), and NPV rate was 85% (95% CI: 74% to 92%), confirming the value of CEUS for detecting lymphoma in the spleen. Additionally, the concordance in the classification of positive findings for benignity between the reference standard and CEUS scans was excellent: 100% of agreement on the spleen benign status (Table 2). Finally, the overall diagnostic accuracy rate of the CEUS test was 96% (95% CI: 93% to 98%), i.e., the results were accurate in 250 of 260 nodules (Table 3).

### 3.3. Distribution of CEUS Perfusional Patterns within the Three Clinical Categories of Lymphomas

To classify the lymphomatous lesions of the spleen within further three DC, i.e., a-NHL-DC, c-HL-DC, and i-NHL-DC, we compared the findings of the wash-out patterns among the 122 a-NHL nodules, 60 c-HL nodules, and 12 i-NHL nodules which were detected at CEUS examinations.

a-NHL nodules displayed isoenhancement in the arterial phase followed by wash-out appearance that had early onset and marked degree within <90 s following contrast agent injection, in a total of 122/122 (100%) nodules (Figure 2). The wash-out onset of the 122 nodules of exploratory a-NHL-DC occurred at a median of 42 s (range 20–55 s) with a median of marked wash-out of 74 s (range 55–90 s).

C-HL nodules showed isoenhancement in the arterial phase followed by early wash-out onset appearance and marked in degree between 90 and 110 s following contrast agent injection, in a total of 57/60 (95%) nodules (Figure 3). The 60 nodules of exploratory c-HL-DC had the onset of wash-out occurring at a median of 43 s (range 25–48 s) with a median of marked wash-out of 100 s (range 92–120 s).

Finally, i-NHL nodules displayed isoenhancement in the arterial phase followed by wash-out appearance that was early in onset and marked in degree between >110 and ≤120 s following contrast agent injection, in a total of 10/12 (83%) nodules (Figure 4). The wash-out onset for the 12 nodules of exploratory i-NHL-DC occurred at a median of 44 s (range 41–59 s) with a median of marked wash-out of 114.5 s (range 101–120 s). 

By focusing on the degree of contrast intensity during the parenchymal-venous phase, the average timing of marked wash-out of nodules was significantly shorter for a-NHL-DC than for c-HL-DC (*p* < 0.0001), as well as being significantly shorter for c-HL-DC than for i-NHL-DC (*p* < 0.0001; Figure 5).

### 3.4. Features of Tumoral Angiogenesis of Splenic Nodules

For several reasons, only in a minority of cases the direct tissue characterization of spleen nodular lesions was available, i.e., the histologic specimen of the target lesion by surgical resection (splenectomy) or radiologically guided CNB. By immunohistochemical and morphometric evaluations, the MVD was investigated in 15 splenic nodules of a-NHL (DLBCL, seven lesions), c-HL (five lesions), and i-NHL (three lesions). Overall, in our post hoc analysis, the grade of MVD was higher for a-NHL nodules, and MVD seemed to be higher also in c-HL nodules. In fact, MVD decreased in the nodules in the following order: mean microvessel count of 30/0.5 mm^2^ for a-NHL, 26/0.5 mm^2^ for c-HL, and 20/0.5 mm^2^ for i-NHL.

## 4. Discussion

The clinical importance of diagnosing splenic involvement by lymphoma has been underlined in several reports: infiltration is considered an adverse prognostic factor [3,4,5,6,7]. Its identification promptly triggers a change in clinical management: patients with indolent lymphomas move from a “wait & watch approach” to therapy start [3,4], and patients with more aggressive lymphomas move from standard therapy to intensified therapy [5,6,18]. Since splenomegaly may occur even if the spleen is not involved by the disease (hyperplastic or congestive enlargement) and the involvement does not necessarily imply spleen enlargement [2], the most cost-effective diagnostic work-up to detect nodular infiltration in this organ must be sought. To the best of our knowledge, this is the first study providing an US imaging algorithm validated to establish the etiology of a large series of splenic nodules at risk of lymphoma, allowing a broad evaluation of differential diagnoses by the CEUS modality itself.

For the first time, based on a robust sample size of 260 nodules (with final diagnoses uniformly controlled by at least one conventional imaging tool and/or clinical laboratory imaging follow-up, in the cases not directly biopsy-proven) we showed that CEUS was able to achieve the diagnosis of splenic malignant lymphoma with excellent accuracy. 

The hallmark sign of lymphomatous nodular infiltration of spleen consisted of arterial phase isoehnacement followed by early (<60 s) and marked (≤120 s) wash-out appearance of sulfur exafluoride microbubbles. This finding validated the effectiveness of temporal CEUS enhancement and wash-out criteria to establish the diagnosis of involvement by lymphoma. The three principal clinical subtypes of lymphomas analyzed in the study, i.e., a-NHL, c-HL, and i-NHL, were differentiated on the basis of the degree of wash-out intensity (from contrast injection) of nodules. The totality of a-NHL nodules was categorized according to the typical wash-out appearance of very marked degree, i.e., within <90 s following contrast agent injection (Figure 2A–C) resulting in a PPV of 100% (95% CI, 97–100%). Most c-HL nodules tended to be categorized in wash-out appearance of discretely marked degree, i.e., between 90 and 110 s following contrast agent injection (Figure 3A–C), with PPV of 96.6% (95% CI, 88.3–99.6%). Most i-NHL nodules tended to segregate in a diagnostic category characterized by a wash-out appearance that was sufficiently marked in degree, i.e., between >110 and ≤120 s following contrast agent injection (Figure 4A–C), with PPV of 77% (95% CI, 46.2–95%). 

In our post hoc analysis regarding tumoral neoangiogenesis evaluation, MVD decreased in the nodules in the following order: mean microvessel count of 30/0.5 mm^2^ for a-NHL, 26/0.5 mm^2^ for c-HL, and 20/0.5 mm^2^ for i-NHL. Abundant angiogenesis such as vessel proliferation (endothelial cell migration and proliferation) and abnormal vascularization (tube formation with stenosis, occlusion, and/or dilation and/or arteriovenous shunts) is recognized as being critical for lymphoma pathogenesis [24]. CEUS features of focal lesions arising in the spleen of patients with lymphoma make the parenchymal phase of the perfusional study fundamental in distinguishing involvement by lymphoma. During this phase, splenic nodules appear as clearly circumscribed defects of enhancement due to rapid wash-out of microbubbles as compared with the homogeneously enhanced normal parenchyma [6]. This was likely due to the neoangiogenic vascular structure of lymphomatous lesions, characterized by abnormal endothelial layer with large fenestration and arteriovenous shunts. Thus, we speculated that angiogenesis played a differential role in the various subtypes of lymphomas of spleen: aggressive lymphomas expressed the more marked wash-out, having the highest neoplastic microvessel density. 

In our hands, CEUS allowed a correct assessment of spleen status providing prompt characterization of nodule(s) in most cases. Our study provides important information on the patients who may benefit most from CEUS exam in the real-life. Out of 260 nodules at risk of lymphoma, 56 (21%) were finally found to be benign and these were all correctly identified by CEUS. This leads to three main considerations. First, in patients with a diagnosis of lymphoma (on lymph node specimen) and splenic nodular lesions suspected, about a quarter of nodules are nonmalignant. Second, the spleens of the subjects (in our series, they were about one-third of the total number of cases analyzed) who carried such nodules were most likely not involved by disease. Finally, in such instances, CEUS can accurately define the benignity status of spleen avoiding more invasive, complex, and dangerous imaging procedures.

However, our study suffers from four major limitations. First, this trial was conducted in a single center; therefore, studies from other institutions are needed to assess (i) interobserver and interequipment ultrasonographic finding variability; and (ii) reproducibility quality of the CEUS reporting a-NHL, c-HL, and i-NHL diagnostic categories, i.e., concordance among sonographers in diagnosing and subtyping spleen lymphomas. Second, a bias error could have been committed due to the retrospective analysis of the study. For example, the small number of nodules due to i-NHL does not allow to draw a conclusion in this subsetting of lymphoma. Thus, it is desirable to conduct prospective studies from multiple institutions. Third, this study does not shed more light on the involvement of spleen by lymphoma, without nodules, because this category was not included in the study. Lastly, this study does not include findings on the metastatic invasion of spleen by solid cancers, i.e., CEUS behaviors of non-hematologic malignant lesions, such as colorectal adenocarcinoma or melanoma. 

## 5. Conclusions

In conclusion, modern ultrasonographic technology is cost-effective as part of the diagnostic work-up of spleen status in patients with lymphomas. We present important indications of CEUS for assessing the spleen at risk of lymphomatous invasion. CEUS can be used as a prompt, harmless, and simple first-step to better characterize nodule(s) detected on routine imaging surveillance at the time of the staging process following lymph node biopsy-proven lymphomas, and during follow-up of previously treated lymphoma. 

New guidelines could therefore reconsider the most appropriate position of CEUS in the diagnostic flow chart of spleen status in patients at risk of lymphomas.

## Figures and Tables

**Figure 1 cancers-14-01927-f001:**
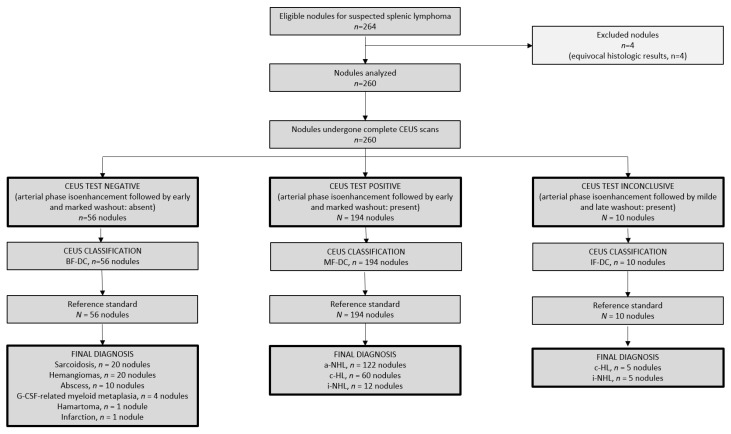
Consort diagram. The flow of nodules through the study. CEUS, contrast-enhanced ultrasonography; BF-DC, benign findings-diagnostic criteria; MF-DC, malignant findings-diagnostic criteria; IF-DC, inconclusive findings-diagnostic criteria; G-CSF, granulocyte-colony stimulating factor; a-NHL, aggressive non-Hodgkin lymphoma; c-HL, classic Hodgkin lymphoma; i-NHL, indolent non-Hodgkin lymphoma. a-NHL included diffuse large B-cell lymphomas (*n* = 108 nodules), T-rich B cell lymphomas (*n* = 10 nodules) and anaplastic T-cell lymphoma (*n* = 4 nodules). c-HL included nodular sclerosis (*n* = 50 nodules), mixed cellularity (*n* = 11 nodules) and lymphocyte-rich (*n* = 4 nodules) subtypes. i-NHL included grade 1–2 follicular lymphoma (*n* = 17 nodules).

**Figure 2 cancers-14-01927-f002:**
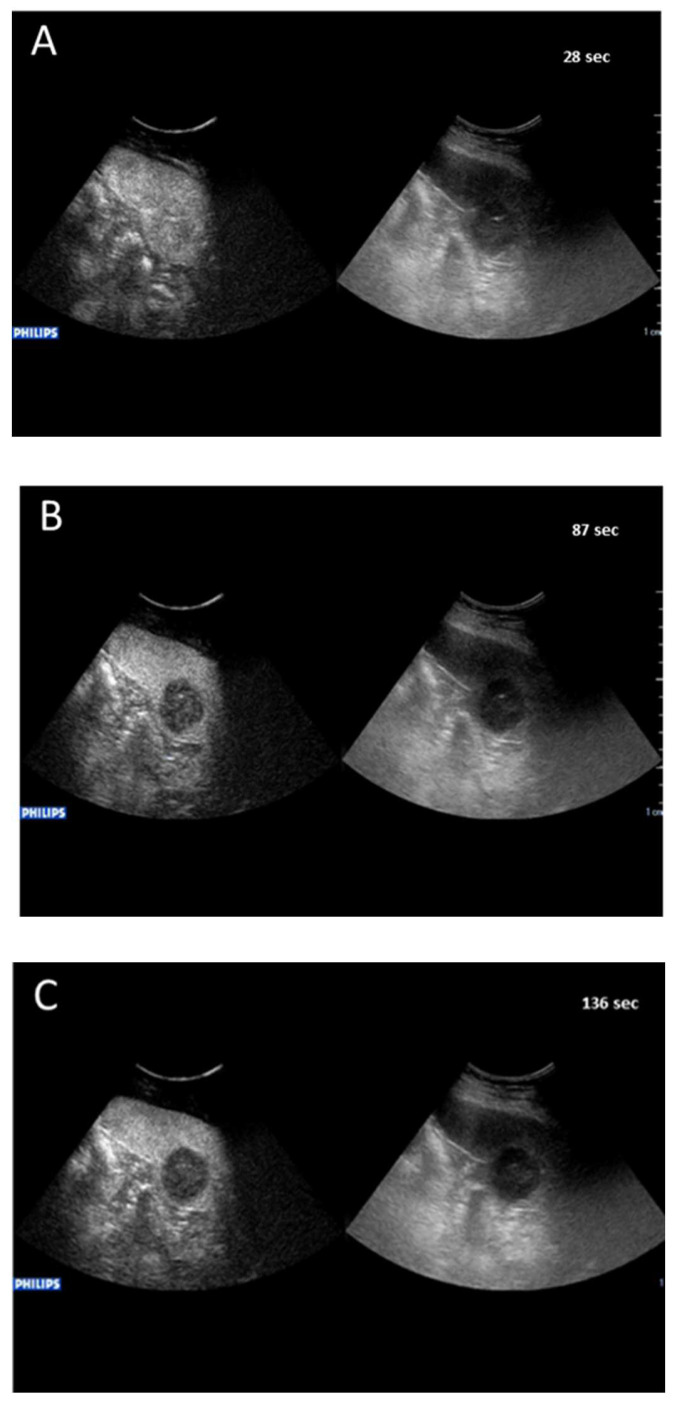
Major features of CEUS according to the exploratory diagnostic criteria. (**A**–**C**) showing aggressive non-Hodgkin lymphoma nodule with very marked wash-out, at 87 s from contrast agent injection.

**Figure 3 cancers-14-01927-f003:**
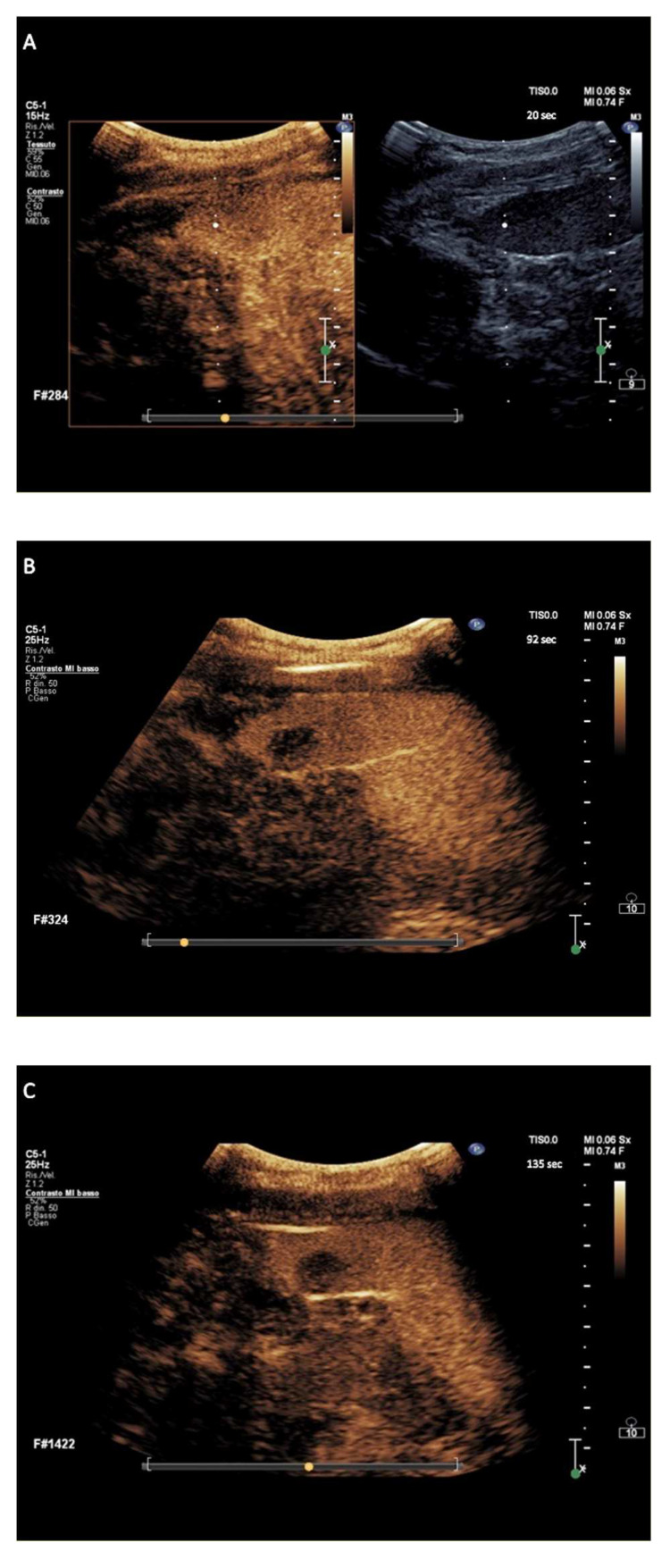
Major features of CEUS according to the exploratory diagnostic criteria. (**A**–**C**) showing classic Hodgkin lymphoma nodule with discretely marked wash-out, at 92 s from contrast agent injection.

**Figure 4 cancers-14-01927-f004:**
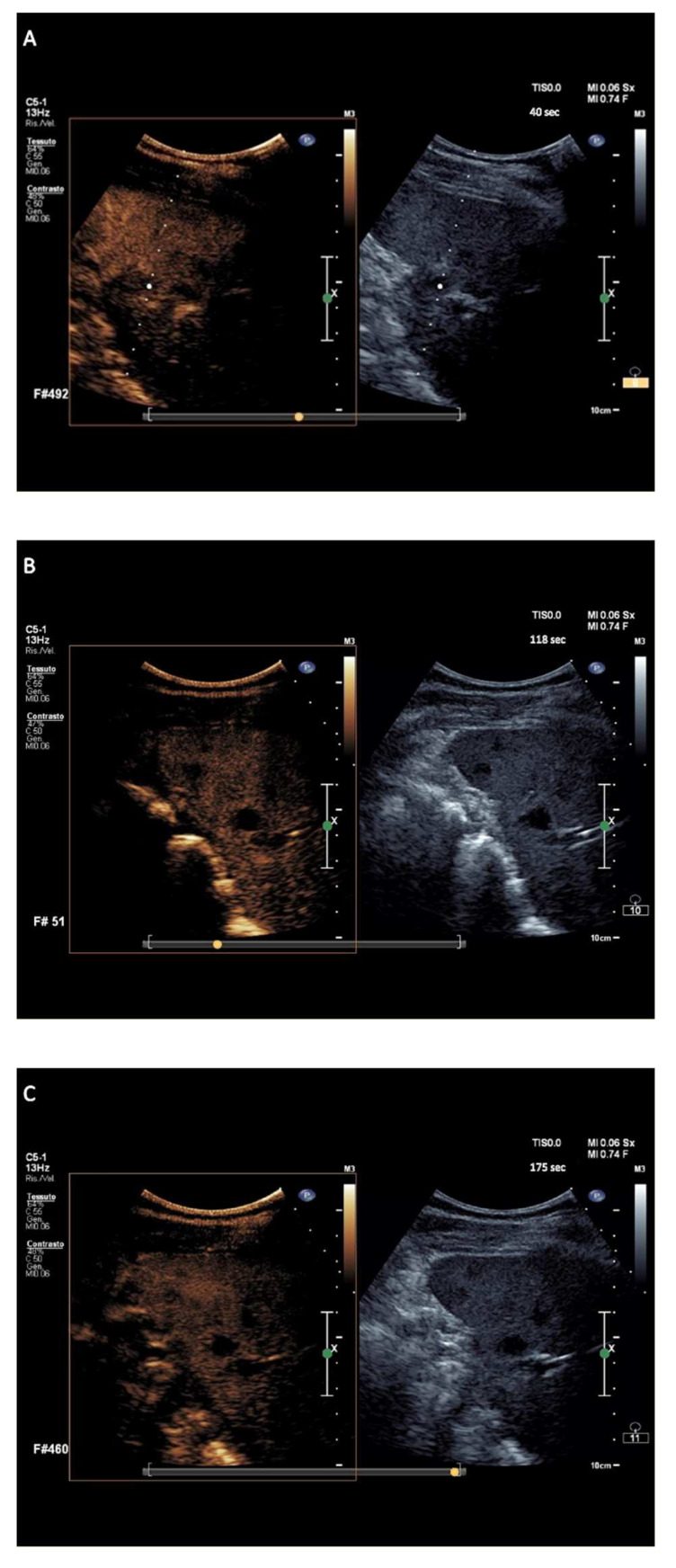
Major features of CEUS according to the exploratory diagnostic criteria. (**A**–**C**) showing indolent non-Hodgkin lymphoma nodules with sufficiently marked wash-out, at 118 s from contrast agent injection.

**Figure 5 cancers-14-01927-f005:**
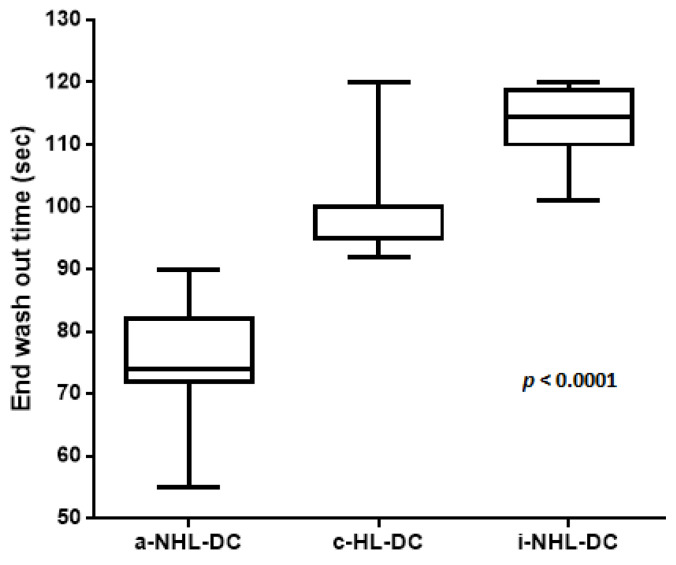
The degree of the contrast intensity of the wash-out of nodules in the three clinical categories of lymphomas. a-NHL-DC, aggressive non-Hodgkin lymphoma diagnostic criteria (122 nodules); c-HL-DC, classic Hodgkin lymphoma diagnostic criteria (60 nodules); i-NHL-DC, indolent non-Hodgkin lymphoma diagnostic criteria (12 nodules).

**Table 1 cancers-14-01927-t001:** Patient and splenic nodules characteristics.

Variable	Patients (*n* = 77)	Nodules (*n* = 260)
Patients’ gender: male	43 (57%)	
Patients’ age: median, years (range)	48 (22–72)	
Lymph node biopsy-proven lymphoma		
Non-Hodgkin lymphoma	44 (57%)	
Hodgkin lymphoma	33 (43%)	
Timing of the discovery of nodules		
Staging for lymphoma	56 (72%)	
Follow-up for lymphoma	21 (28%)	
Anti-lymphomatous therapy administered *		
R-CHOP-21	37 (48%)	
CHOP-21	4 (5%)	
R-Bendamustin	3 (4%)	
ABVD	33 (43%)	
Nodule size: median, cm (range)		1.5 (0.5–7)
Aetiology of nodules		
Malignancy **		204 (78%)
a-NHL		122 (60%)
DLBCL		108 (53%)
TRBCL		10 (5%)
ATCL		4 (2%)
c-HL		65 (32%)
NS-HL		50 (25%)
MC-HL		11 (5%)
LR-HL		4 (2%)
i-NHL		17 (8%)
FL		17 (8%)
Benignity ***		56 (22%)
Sarcoidosis		20 (36%)
Hemangiomas		20 (36%)
Abscess		10 (17%)
Myeloid metaplasia		4 (7%)
Hamartoma		1 (1%)
Splenic infarction		1 (1%)

Note: a-NHL, aggressive non-Hodgkin lymphoma: DLBCL (diffuse large B cell lymphoma), TRBCL (T-rich B cell lymphoma), and ATCL (anaplastic T-cell lymphoma); c-HL, classic-Hodgkin lymphoma: NS-HL (nodular sclerosis Hodgkin lymphoma), MC-HL (mixed cellularity Hodgkin lymphoma), and LR-HL (lymphocyte-rich Hodgkin lymphoma); i-NHL, indolent non-Hodgkin lymphoma: FL (follicular lymphoma, grade 1–2). * Therapy for treating the underlying lymphomas was scheduled according to the National Comprehensive Cancer Network (NCCN) guidelines (ref #18). ** The definitive diagnoses of lymphomatous nodules were established on the basis of histologic results of superficial or deep-seated lymph node biopsies plus clinical/laboratory/imaging follow-up for 174 nodules, and splenectomy or radiologically guided core needle biopsy for 30 nodules. *** The definitive etiologies of benign lesions were performed by clinical/laboratory/imaging follow-up for 46 nodules, and blood culture microbiological results and subsequent specific anti-microbial therapy for 10 nodules.

**Table 2 cancers-14-01927-t002:** Contrast-enhanced ultrasonography (CEUS) dynamic patterns of the benign nodules.

Aetiology	Median Size, cm (Range)	CEUS Patterns		Reference Standard
		Arterial Phase	Portal Phase	
Sarcoidosis, n = 20	1.5 (0.5–2.2)	Globally, hypoechoic compared to the surrounding parenchyma	Globally, hypoechoic (without later washout) compared to the surrounding parenchyma	Lymph nodes biopsy plus imaging follow-up after treatment, *n* = 20
Hemangiomas, *n* = 20	1.5 (1.1–6)	Isoenhancement (*n* = 10), capillary hemangiomas Discontinuous globular peripheral enhancement (*n* = 10), cavernous hemangiomas	With or without later washout of mild degree as intensity (*n* = 10), capillary hemangiomas Progressive centripetal fill-in or very late (>3 min) washout of mild degree (*n* = 10), cavernous hemangiomas	Clinical and imaging follow-up, *n* = 20
Abscesses, *n* = 10	1.1 (0.7–1.5)	Thin rim hyperenhancement pattern	Washout	Blood culture, *n* = 10 (positive test for fungus, *n* = 8; positive test for bacteria, *n* = 2) plus imaging monitoring after anti-microbial therapy
G-CSF-related myeloid metaplasia, *n* = 4	2.5 (2–3.2)	Relative hypoenhancement of the periphery of nodule and normal enhancement in the remaining portion of lesion compared to the surrounding parenchyma (reversed rim-enhancement)	Isoenhancement	Clinical and imaging follow-up, *n* = 4
Hamartoma, *n* = 1	7	Hyperenhancement	Hyperenhancement	Clinical and imaging follow-up, *n* = 1
Infarction, *n* = 1	6	Hypo-enhancement (triangular-shaped)	Hypo-enhancement (triangular-shaped)	Clinical and imaging follow-up, *n* = 1

Note: G-CSF, granulocyte colony-stimulating factor.

**Table 3 cancers-14-01927-t003:** Accuracy of CEUS for the diagnosis of spleen involvement by lymphoma on 260 nodules analyzed in the study.

Accuracy Measurement	Results
Reference standard *	100% (260/260 nodules)
Sensitivity	95% (95% CI, 91–98)
Specificity	100% (95% CI, 94–100)
Positive predictive value	100% (95% CI, 98–100)
Negative predictive value	85% (95% CI, 74–92)
False-negative finding	5% (10/260 nodules)
False-positive finding	–
Overall diagnostic accuracy	96% (95% CI, 93–98)

Note: CEUS, contrast-enhanced ultrasonography; CI, confidence interval. * The definitive diagnoses of lymphomatous nodules were established on the basis of histologic results of superficial or deep-seated lymph node biopsies plus clinical/laboratory/imaging follow-up for 174 nodules, and splenectomy or radiologically guided core needle biopsy for 30 nodules. The definitive etiologies of benign lesions were performed by clinical/laboratory/imaging follow-up for 46 nodules, and blood culture microbiological results and subsequent specific anti-microbial therapy for 10 nodules.

## Data Availability

The data presented in this study are available on request from the corresponding author. The data are not publicly available due to privacy restrictions.

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
