# Peer review of "Sulfur Exafluoride Contrast-Enhanced Ultrasound Showing Early Wash-Out of Marked Degree Identifies Lymphoma Invasion of Spleen with Excellent Diagnostic Accuracy: A Monocentric Study of 260 Splenic Nodules"

_cancers, 2022, doi:10.3390/cancers14081927_

Round 1

Reviewer 1 Report

This study demonstrates that CEUS is effective in the identification of splenic involvement by lymphoma in patients undergoing clinical staging or post-treatment follow up.  It shows high sensitivity, specificity and positive predictive value for the diagnosis of lymphoma.  Furthermore, aggressive NHL, CHL and indolent NHL demonstrate different CEUS perfusion patterns which can be used to differentiate these entities.  If these results can be confirmed by multicenter and prospective studies, CEUS will become a cost-effective tool in the clinical management of patients with lymphomas.

Comments:

  1. This cohort includes only the common lymphoma types.  Interestingly, all the indolent lymphoma cases are grade 1-2 follicular lymphoma.  Is there an explanation for that?  Since only a small subset of cases had histologically confirmed diagnosis. the classification may not be accurate in some cases.
  2. Is there any difference between histologically confirmed cases and the rest of case with regards to diagnostic sensitivity, specificity and positive predictive value?

Author Response

Your comment is very punctual, and we’ve raised the same consideration regarding classification accuracy. Although histologic evidence is the reference standard for the diagnosis of involvement by lymphoma, for obvious practical and ethical reasons, it is not possible to perform biopsies in all organs of interest, especially the spleen. In the discussion we considered the small number of nodules of i-NHL as a study limitation since it doesn’t allow to draw a conclusion in this sub-setting of lymphomas. Regarding the second comment of the same reviewer, in the small subgroup of patients who underwent nodules biopsy or splenectomy the histologic results did not modify the CEUS diagnostic accuracy

Reviewer 2 Report

The authors showed the usefulness of CEUS for diagnosing lymphoma. This study is important important for diagnosis. I have one question. In the case of worse progress, how would the image of CEUS change ?

Author Response

The question raises an interesting point but unfortunately, we don’t have the data yet. It could be a great inspiration for a follow up study in the same cohort.